# A clinical prediction model to identify children at risk for revisits with serious illness to the emergency department: A prospective multicentre observational study

Ruud G. Nijman[1,2☯]*, Dorine H. Borensztajn[3☯], Joany M. Zachariasse[3☯], Carine Hajema[3], Paulo Freitas[4], Susanne Greber-Platzer[5], Frank J. Smit[6], Claudio F. Alves[7], Johan van der Lei[8], Ewout W. Steyerberg[9], Ian K. Maconochie[2], Henriette A. Moll[3]

1 Department of Infectious Diseases, Section of Paediatric Infectious Diseases, Imperial College of Science, Technology and Medicine, Faculty of Medicine, London, United Kingdom, 2 Department of Paediatric Emergency Medicine, St Mary's Hospital–Imperial College NHS Healthcare Trust, London, United Kingdom, 3 Department of General Paediatrics, Erasmus MC-Sophia Children's Hospital, Rotterdam, The Netherlands, 4 Intensive Care Unit, Hospital Prof. Dr. Fernando Fonseca, Lisbon, Portugal, 5 Department of Paediatrics and Adolescent Medicine, Medical University Vienna, Vienna, Austria, 6 Department of Paediatrics, Maasstad Hospital, Rotterdam, The Netherlands, 7 Department of Paediatrics, Hospital Prof. Dr. Fernando Fonseca, Lisbon, Portugal, 8 Department of Medical Informatics, Erasmus MC- University Medical Centre Rotterdam, Rotterdam, The Netherlands, 9 Department of Medical Statistics and Bioinformatics, Leiden University Medical Centre, Leiden, The Netherlands

☯ These authors contributed equally to this work.
* r.nijman@imperial.ac.uk

## Abstract

### Background

To develop a clinical prediction model to identify children at risk for revisits with serious illness to the emergency department.

### Methods and findings

A secondary analysis of a prospective multicentre observational study in five European EDs (the TRIAGE study), including consecutive children aged <16 years who were discharged following their initial ED visit ('index' visit), in 2012–2015. Standardised data on patient characteristics, Manchester Triage System urgency classification, vital signs, clinical interventions and procedures were collected. The outcome measure was serious illness defined as hospital admission or PICU admission or death in ED after an unplanned revisit within 7 days of the index visit. Prediction models were developed using multivariable logistic regression using characteristics of the index visit to predict the likelihood of a revisit with a serious illness. The clinical model included day and time of presentation, season, age, gender, presenting problem, triage urgency, and vital signs. An extended model added laboratory investigations, imaging, and intravenous medications. Cross validation between the five sites was performed, and discrimination and calibration were assessed using random effects models. A digital calculator was constructed for clinical implementation. 7,891 children out

**Data Availability Statement:** All data are available from https://data.hpc.imperial.ac.uk/resolve/?doi= 8375&access=.

**Funding:** No specific funding was received for this study. RN was supported with a Thrasher Research Fund grant (Award number 12830) and was awarded an NIHR academic clinical lectureship (CL-2018-21-007) to do this work. The funders had no role in study design, data collection and analysis, decision to publish, or preparation of the manuscript.

**Competing interests:** The authors have declared that no competing interests exist.

**Abbreviations:** aOR, adjusted Odds Ratio; AUC, Area Under the receiving operating characteristic Curve; AVPU, Alert-Verbal-Pain-Unresponsive; CI, confidence interval; ED, emergency department; IQR, interquartile range; Iv, intravenous; LR, likelihood; MTS, Manchester triage system; OR, odds ratio; PICU, paediatric intensive care unit; VIF, variance inflation factor.

of 98,561 children had a revisit to the ED (8.0%), of whom 1,026 children (1.0%) returned to the ED with a serious illness. Rates of revisits with serious illness varied between the hospitals (range 0.7–2.2%). The clinical model had a summary Area under the operating curve (AUC) of 0.70 (95% CI 0.65–0.74) and summary calibration slope of 0.83 (95% CI 0.67–0.99). 4,433 children (5%) had a risk of > = 3%, which was useful for ruling in a revisit with serious illness, with positive likelihood ratio 4.41 (95% CI 3.87–5.01) and specificity 0.96 (95% CI 0.95–0.96). 37,546 (39%) had a risk <0.5%, which was useful for ruling out a revisit with serious illness (negative likelihood ratio 0.30 (95% CI 0.25–0.35), sensitivity 0.88 (95% CI 0.86–0.90)). The extended model had an improved summary AUC of 0.71 (95% CI 0.68–0.75) and summary calibration slope of 0.84 (95% CI 0.71–0.97). As study limitations, variables on ethnicity and social deprivation could not be included, and only return visits to the original hospital and not to those of surrounding hospitals were recorded.

## Conclusion

We developed a prediction model and a digital calculator which can aid physicians identifying those children at highest and lowest risks for developing a serious illness after initial discharge from the ED, allowing for more targeted safety netting advice and follow-up.

## Introduction

A small, but not negligible proportion of children (2.2% to 7%) will return to the emergency department (ED) within the course of their illness after an initial discharge from the ED [1–6]. Rates of children with a revisit to the ED increased in recent years, partly contributing to the overcrowding of emergency departments [5]. Until now, most interest has been in understanding revisits which do not require emergency care, and avoiding those revisits that could be managed safely in the community or with appropriate safety netting information. Several factors at the time of an initial ED visit, such as young age, nighttime presentation and high acuity level, have been linked to an increased risk of having a revisit to the ED [3, 4, 7–10]. Yet, less is known about children returning to the ED with a serious illness, seen in 13 to 42% of the children with an ED revisit [3, 4, 7, 9, 11].

At present, no prospective study has looked at determinants associated with return visits in children in EDs across Europe, let alone with a focus on return visits with serious illness. Increasing our understanding of determinants associated with return visits to EDs with serious illness will aid the identification of patients that can be discharged safely with a low risk for clinical deterioration, allowing for improved ED discharge flow and alleviating ED crowding. It will also create opportunities for more targeted safety netting and follow up for those patients at a higher risk. Hence, we aimed to develop a clinical prediction model and digital calculator to support clinicians identifying children at high and low risks of a return visit with serious illness.

## Methods

We first looked at the number of return visits with a serious illness amongst children who had previously presented to the ED in five hospitals across Europe. We then looked at characteristics of the initial ('index') ED visit and the association with a return visit with a serious illness. Finally, we developed and validated a clinical prediction to estimate the risk of an individual

patient to return to the ED within seven days of initial ED discharge, following TRIPOD guidelines for the development and validation of clinical prediction models.

## Design, participants and setting

The study was embedded in the TrIAGE (TRiage Improvements Across General Emergency departments) project, aiming to improve the early recognition of seriously ill children at the ED [12, 13]. We performed a secondary analysis of data collected as part of a prospective observational multicentre cohort study in five European EDs (Erasmus MC, Rotterdam, the Netherlands (EMC); Maasstad Hospital, Rotterdam, the Netherlands (MHR); St Mary's Hospital, London, United Kingdom (SMH); Hospital Fernando da Fonseca, Lisbon, Portugal (HFF); Medizinische Universitaet Wien, Vienna, Austria (MUW)), with data collected between 2012 and 2015 (Table 1). Data analyses were planned and executed following completion of the original data collection. Consecutive children aged 0 to 16 years who presented to EDs for unscheduled healthcare were included in the study. Universal health care coverage was available across the participating sites, with similar and free of charge access to urgent and emergency care, and primary care. Primary care with either general practitioners and/or primary care paediatricians was available in all the countries in our study. Patients who were redirected towards a primary care provider or urgent care centre after presenting to the ED at the time of their index visit were not eligible for this study, as we did not record complete clinical data for these children. Patients who left against medical advice or who left before being seen were only eligible if sufficient baseline data were available for that visit. Children who were admitted following their index visit were subsequently excluded from analysis, as they have a different risk profile for a revisit with serious illness [5, 14]. A waiver for the need of patient informed consent was obtained at all participating hospitals.

## Data collection and predictor variables

In all hospitals data on: patient characteristics, triage urgency, presenting problem, vital signs, ED interventions and procedures, treatment, and outcomes were recorded in the electronic health records in a standardised manner as part of routine clinical practice. Predictor variables were chosen based on their known association with return visits, patient urgency and severity scores [15, 16]. Data were extracted from the hospital's electronic systems and checked for completeness, validity and outliers. Season, time and day of presentation were derived from the patient's arrival date and time. Seasons were divided in Winter (21/12–20/03), Spring (21/

**Table 1. Overview of the participating hospitals.**

| | Erasmus MC, Rotterdam, the Netherlands | Maasstad Hospital, Rotterdam, the Netherlands | St Mary's Hospital, London, United Kingdom | Hospital Fernando da Fonseca, Lisbon, Portugal | Medizinische Universitaet Wien, Vienna, Austria |
|---|---|---|---|---|---|
| **Hospital characteristics** | University hospital | District general hospital | University hospital | District general hospital | University hospital |
| | 60 paediatric beds | 59 paediatric beds | 46 paediatric beds | 91 paediatric beds | 134 paediatric beds |
| **Catchment area** | Urban | Urban | Urban | Mixed urban and rural | Urban |
| | Mixed high and low socio-economic status | Generally low socio-economic status | Mixed high and low socio-economic status | Generally low socio-economic status | Mixed high and low socio-economic status |
| **ED characteristics** | Paediatrics | Mixed adult-paediatric | Paediatrics | Paediatrics | Paediatrics |
| | 6500 children/year | 9500 children/year | 27,000 children/year | 60,000 children/year | 22,000 children/year |
| **Supervising physician** | Paediatrician | Paediatrician | Paediatric emergency medicine physician | Paediatrician | Paediatrician |
| **Inclusion period** | 01-01-2012 to 31-12-2014 | 01-05-2014 to 31-10-2015 | 01-07-2014 to 28-02-2015 | 01-03-2014 to 28-02-2015 | 01-01-2014 to 31-12-2014 |

03–20/06), Summer (21/06–20/09), Autumn (21/09–20/12). Date and time of presentation were grouped into: Weekday evenings (Monday–Friday; 18.00–23.00 hrs), Weekday nights (Monday–Friday; 23.00–07.00 hrs), Weekend days (Saturday–Sunday; 08.00–18.00 hrs), Weekend evenings (Saturday–Sunday; 18.00–23.00 hrs), Weekend nights (Saturday–Sunday; 23.00–07.00 hrs), and Weekdays (Monday–Friday; 08.00–18.00 hrs). Triage urgency was determined using the third edition of the Manchester Triage System [17]. This is a validated 5-level urgency classification system that allocates an urgency category using specific discriminators for 52 flowcharts detailing varying presenting problems [18]. These flowcharts were re-classified into 11 main groups of presenting symptoms (S1 Table). Low urgency levels standard (level 4) and non-urgent (level 5), as well as emergent (level 1) and very urgent (level 2) were combined in the multivariable regression analyses due to low number of cases in the highest and lowest category. Vital signs were measured at the time of triage by the triage nurse, following advanced paediatric life support standards [19]. Heart rate (beats per minute) and respiratory rate (breaths per minute) were categorised using the age-specific Advanced Paediatric Life Support threshold values. Oxygen saturations were dichotomised at <94%; body temperature at > = 38.0 degrees Celsius. For level of consciousness, Verbal, Pain, and Unconsciousness on the Alert-Verbal-Pain-Unresponsive (AVPU) scale were combined into 'reduced level of consciousness'. Interventions and procedures in the ED included: any laboratory tests performed (ie. blood, urine, cerebral spinal fluid, other), any imaging done (ie. Xray, ultrasound, CT, MRI), and any intravenous medications and/or fluids administered. Immediate lifesaving interventions included airway and breathing support, haemodynamic support, emergency procedures, and emergency medications (S1 Table). Blood pressure, capillary refill and lifesaving interventions were not used for prediction modelling owing to low number of values measured in children across all institutions (blood pressure available for 8%; capillary refill for 47%); and/or <10 cases with abnormal values for children with serious illness at return visit (Table 2). In three hospitals (EMC, SMH, MUW), co-morbidity was coded into complex and non-complex by trained medical students blinded for the outcome of this study according to the definitions by Simon et al. [20].

## Outcome measures

Serious illness was defined as the need for hospital admission or paediatric intensive care unit (PICU) admission or death in ED after an unplanned return visit within 7 days of the index visit. Characteristics of children admitted to PICU at the time of their revisit were described in detail separately (S2 Table).

## Statistical analysis

Firstly, we performed uni- and multi-covariate logistic regression to look at the association between the characteristics of the index visit and a return visit with a serious illness, reporting crude unadjusted and adjusted odds ratios (OR) with 95% confidence intervals. For the three hospitals (Erasmus cohort, London cohort, Vienna cohort) with available data on co-morbidity we performed an additional analysis to determine the role of co-morbidity on return visits with serious illness in these institutions. Our cohort provided sufficient power for both the derivation and cross-validation studies, estimating the required sample using the recommended approach by Riley et al. (S1 Appendix) [21–23].

Secondly, we developed two clinical prediction models using logistic regression. A first model included clinical characteristics of the index visit (ie: 'clinical model'), and then the use of laboratory investigations, intravenous medications or fluids, and imaging were added to the model (ie.: 'extended model') as a second step. We included the five hospitals as a categorical

**Table 2. Characteristics of children at the time of their index visit who had and did not have a revisit with serious illness (n = 98,561).**

| | All children with index visit | Children without serious illness at revisit | Children with serious illness at revisit | OR [+] (95% CI) |
|---|---|---|---|---|
| *Total n (%)* | *98,561 (100%)* | *97,535 (99%)* | *1,026 (1.0%)* | |
| **Hospital ^** | 98,561 (100%) | | | |
| Erasmus MC | 13,318 (13%) | 12,855 (13%) | 283 (28%) | 3.18 (2.58–3.92) |
| Maasstad Hospital | 7,650 (8%) | 7,491 (8%) | 159 (16%) | 3.07 (2.43–3.88) |
| St Mary's Hospital | 12,986 (13%) | 12,871 (13%) | 115 (11%) | 1.29 (1.00–1.66) |
| Hospital Fernando da Fonseca | 46,013 (47%) | 45,673 (47%) | 340 (33%) | 1.08 (0.88–1.32) |
| Medizinische Universitaet Wien | 18,774 (9%) | 18,645 (19%) | 219 (13%) | *Reference* |
| **Gender ^** | 98,561 (100%) | | | |
| Female | 45,595 (46%) | 45,148 (46%) | 447 (44%) | 0.90 (0.79–1.01) |
| **Age ^** | 98,561 (100%) | | | |
| Median (IQR) | 4.7 (1.9–9.7) | 4.7 (1.9–9.7) | 2.5 (0.8–7.1) | - |
| <1 year | 13,715 (14%) | 13,415 (14%) | 300 (29%) | 2.98 (2.40–3.70) |
| 1 - <2 years | 12,213 (12%) | 12,061 (12%) | 152 (15%) | 1.68 (1.31–2.14) |
| 2 - <5 years | 25,314 (26%) | 25,083 (26%) | 231 (23%) | 1.23 (0.98–1.54) |
| 5 - <12 years | 32,160 (33%) | 31,930 (33%) | 230 (22%) | 0.96 (0.77–1.20) |
| 12 –<16 years | 15,159 (15%) | 15,046 (15%) | 113 (11%) | *reference* |
| **Time and day of presentation ^** | 98,559 (100%) | | | |
| Weekdays | 42,004 (43%) | 41,592 (43%) | 412 (40%) | 0.95 (0.80–1.12) |
| Weekday evenings | 22,150 (23%) | 21,944 (23%) | 206 (20%) | 1.41 (1.14–1.75) |
| Weekday nights | 7,470 (8%) | 7,367 (8%) | 103 (10%) | 1.15 (0.96–1.37) |
| Weekend days | 15,131 (15%) | 14,961 (15%) | 170 (17%) | 1.05 (0.83–1.33) |
| Weekend evenings | 8,224 (8%) | 8,139 (8%) | 85 (8%) | 1.43 (1.06–1.92) |
| Weekend nights | 3,580 (4%) | 3,530 (4%) | 50 (5%) | *reference* |
| **Season ^** | 98,561 (100%) | | | |
| Winter | 25,077 (25%) | 24,784 (25%) | 293 (29%) | 1.23 (1.04–1.45) |
| Spring | 22,813 (23%) | 22,559 (23%) | 254 (25%) | 1.17 (0.98–1.39) |
| Summer | 21,781 (22%) | 21,578 (22%) | 203 (20%) | 0.98 (0.81–1.17) |
| Fall | 28,890 (29%) | 28,614 (29%) | 276 (27%) | *reference* |
| **Flowchart for presenting problem (MTS)^** | 97,279 (99%) | | | |
| Shortness of breath | 10,224 (11%) | 10,037 (10%) | 187 (18%) | 2.96 (2.16–4.06) |
| ENT problems | 9,392 (10%) | 9,359 (10%) | 33 (3%) | 0.56 (0.36–0.87) |
| Gastro-intestinal problems | 14,199 (15%) | 13,953 (15%) | 246 (24%) | 2.80 (2.06–3.81) |
| Neurological problem | 3,136 (3%) | 3,079 (3%) | 57 (6%) | 2.94 (2.01–4.32) |
| Unwell child | 19,375 (20%) | 19,087 (20%) | 288 (28%) | 2.40 (1.77–3.25) |
| Urological problems | 2,154 (2%) | 2,135 (2%) | 19 (2%) | 1.42 (0.83–2.41) |
| Rash | 7,684 (8%) | 7,651 (8%) | 33 (3%) | 0.69 (0.44–1.07) |
| Abscess and soft tissue infection | 1,505 (2%) | 1,484 (2%) | 21 (2%) | 2.25 (1.35–3.76) |
| Trauma | 17,916 (18%) | 17,844 (19%) | 72 (7%) | 0.64 (0.45–092) |
| Wounds | 3,854 (4%) | 3,843 (4%) | 11 (1%) | 0.46 (0.24–0.88) |
| Other | 7,840 (8%) | 7,791 (8%) | 49 (5%) | *Reference* |
| **Triage urgency classification (MTS) ^** | 97,292 (99%) | | | |
| Emergent | 321 (0.3%) | 319 (0.3%) | 2 (0.3%) | *(combined with very urgent)* |
| Very urgent | 7,278 (8%) | 7,120 (7%) | 158 (16%) | 3.28 (2.73–3.94) |
| Urgent | 25,513 (26%) | 25,075 (26%) | 438 (43%) | 2.67 (2.33–3.05) |

*(Continued)*

**Table 2.** (Continued)

| | All children with index visit | Children without serious illness at revisit | Children with serious illness at revisit | OR + (95% CI) |
|---|---|---|---|---|
| Standard | 62,617 (64%) | 62,207 (65%) | 410 (40%) | *Reference (combined standard and non-urgent)* |
| Non-urgent | 1,563 (2%) | 1,555 (2%) | 8 (0.8%) | *Reference (combined standard and non-urgent)* |
| **Tachycardia (beats/minute) ^** | 54,438 (55%) | | | |
| Present, APLS thresholds | 10,463 (19%) | 10,252 (19%) | 211 (30%) | 1.68 (1.45–1.94) |
| **Tachypnoea (breaths/minute) ^** | 45,963 (47%) | | | |
| Present, APLS thresholds | 8,945 (20%) | 8,770 (19%) | 175 (33%) | 1.56 (1.34–1.82) |
| **Systolic blood pressure ^** | 7,867 (8%) | | | |
| Median (IQR), mmHg | 112 (103–120) | 112 (103–119) | 112 (102–122) | + |
| **Capillary refill ^** | 46,748 (47%) | | | |
| > = 3 seconds | 420 (0.9%) | 411 (0.9%) | 9 (1.6%) | + |
| **Temperature (degrees Celsius) ^** | 75,771 (77%) | | | |
| > = 38.0˚C | 9,615 (10%) | 9,426 (10%) | 189 (18%) | 1.81 (1.54–2.12) |
| **Oxygen saturations (%O2) ^** | 52,666 (53%) | | | |
| <94% O2 | 687 (0.7%) | 666 (0.7%) | 21 (2%) | 2.53 (1.63–3.94) |
| **Level of consciousness ^** | 81,551 (83%) | | | |
| Decreased (Verbal–Pain Unresponsive) | 425 (0.5%) | 406 (0.5%) | 19 (2.3%) | 3.84 (2.39–6.17) |
| **Immediate lifesaving interventions^ \*** | 85,575 (87%) | | | |
| Any | 67 (0.1%) | 66 (0.1%) | 1 (0.1%) | + |
| **Laboratory testing at ED ^** | 98,561 (100%) | | | |
| Any | 16,054 (16%) | 15,707 (16%) | 347 (34%) | 2.66 (2.34–3.03) |
| **Imaging at ED ^** | 98,561 (100%) | | | |
| Any | 19,696 (20%) | 19,480 (20%) | 216 (21%) | 1.07 (0.92–1.24) |
| **Medication at ED ^** | 98,561 (100%) | | | |
| Iv medication or iv fluids | 4,295 (4%) | 4,187 (4%) | 108 (11%) | 2.62 (2.14–3.20) |

^ available data, n (%).

\* definition of life saving interventions in S1 Table; airway and breathing support, n = 7; emergency procedures and haemodynamic support, n = 16; emergency medications, n = 23; electrical therapy, n = 0.

+: crude unadjusted ORs based on imputed datasets; Blood pressure not shown because of high % missing values; Capillary refill not shown because of % missing values and <10 cases of serious illness on revisit; immediate lifesaving interventions not included as n<10 for serious illness on revisit.

CI confidence interval; ED: emergency department; MTS: Manchester Triage System; IQR: interquartile range; iv: intravenous.

variable to account for case mix differences between the cohorts. All variables were pre-selected, and variables were included independent from their statistical contribution to the models based on previous literature. No variables were excluded from the final prediction models based on statistical contribution or when adding the variables of the extended model [15]. The overall performance of the extended model was compared with the clinical model using the likelihood ratio test. Missing data for vital signs (Table 2) were imputed using a multiple imputation model including a hospital variable, age, available vital signs, triage urgency, presenting problem, and discharge destination. This resulted in twenty-five iterative databases, including data from all five cohorts, on which we performed statistical analysis after which results were pooled [24]. Imputation was performed using the multivariate imputation by chained equations.

We initially derived the prediction model in four of the five cohorts, and then cross-validated the model in the fifth cohort that was left out. We repeated this "leave-one-out" approach as a method for cross-validating the prediction model four times with each of the five cohorts serving as an independent validation cohort once [25]. Each time, we adjusted the intercept by estimating the linear predictors for the validation cohort without using the hospital variable and then recalculating an intercept, thus acknowledging differences between the hospitals. External validity of the model was determined by estimating the discrimination by means of the Area Under the operating characteristic Curve (AUC), and model calibration by means of calibration plots, the calibration coefficient, or calibration slope, and change in intercept of the model. The AUC is used to estimate the ability to discriminate between patients with and without a return visit with serious illness, with values ranging between 0.5 (indiscriminate test) and 1 (optimal test). Calibration refers to the level of agreement between predicted risks and observed outcome; the ideal calibration coefficient, or slope, being one, indicating perfect agreement between observed and predicted risks. The calibration intercept reflects the extent that predictions are systematically too low or too high ('calibration-in-the-large'), e.g. due to differences in the underlying incidence between cohorts, with an optimal value of zero [26]. The five estimates of AUCs and calibration coefficients of each of the five validation cohorts were then pooled by means of random effects models to provide summary estimates. Finally, we developed a clinical prediction model combining all available data from the five cohorts and with a refitted intercept to allow for hospital differences. An apparent optimism corrected AUC was calculated by means of bootstrapping (500 iterations). We performed bias corrected calibration with 200 bootstrapping iterations to demonstrate goodness of fit for the final overall prediction models. Variance inflation factors were calculated to assess collinearity between predictor variables. Diagnostic performance measures, including sensitivity, specificity and likelihood ratios (LRs) were calculated for different risk thresholds of both the clinical and the extended model. We used R statistical software version 4.0.0 for all our analyses, including use of the rms, metaphor, MICE, pmsampsize, pROC and epiR packages.

### Ethics statement

The study was approved by the participating institutions' medical ethical committees: Medical Ethics Committee Erasmus MC (MEC-2013-567), Maasstad Ziekenhuis Board of Directors (Protocol L2013-103), Imperial College London Joint Research Compliance Office (Reference number: 14SM2164; Ethics reference number 14/WA/1051), Comissão de Ética para a Saúde do Hospital Prof. Dr. Fernando Fonseca EPE (Reunião de 06 de Dezembro de 2017), Ethik Kommission Medizinische der Medizinischen Unversität Wien (EK Nr: 1405/2014). All waived the requirement for informed consent.

## Results

### Rates of revisits

109,482 children with an ED visit were included, of whom 98,561 children (90%) were discharged from ED (Fig 1). A total of 1,026 children (1.0%) returned to the ED with serious illness out of a total of 7,891 children who represented to the ED. Rates of revisits with serious illness varied between the hospitals (range 0.7–2.2%); unadjusted odds ratios for variables of interest and their risk of a return visit with serious illness are shown in Table 2. Twenty-three children (0.02%) were admitted to PICU and one child died at revisit (characteristics of PICU patients presented in S2 Table).

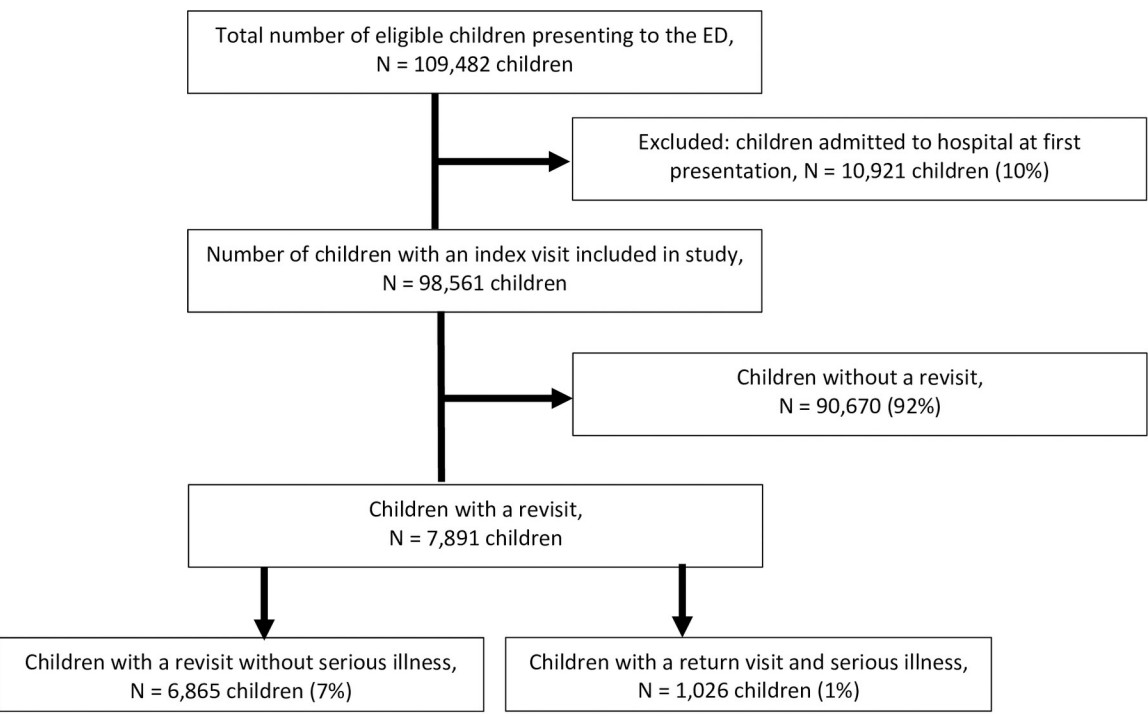

**Fig 1. Flowchart of included population.** Flowchart with exclusion criteria and the included population.

## Characteristics of return visits with serious illness

Several clinical characteristics of the index visit were associated with a revisit with serious illness. These included young age, female gender, time of presentation during the weekend day-time and weekday night-time, and higher triage urgency (Table 3). Presenting problems such as shortness of breath, gastro-intestinal problems, neurological problems and soft tissue infections were also linked to an increased risk of a return visit with serious illness, as were any laboratory tests performed, any imaging done, and any intravenous medications and/or fluids administered. Presentations for trauma and wounds made a revisit with serious illness less likely.

Comorbidity was significantly associated with a return visit with serious illness in the three hospitals for which we had these data, both for children with non-complex (aOR 2.36, 95% CI 1.82–3.05)) and children with complex co-morbidities (aOR 3.39, 95% CI 2.67–4.29) (S3 Table).

## Cross validation of the clinical and extended prediction models

Table 4 presents the overall discrimination and calibration of the "leave-one-out"cross-validation of the clinical and the extended model. Intercept coefficients ranged from -0.91 to 0.10 for the clinical model showing some inter-hospital variation for calibration-in-the-large. Discrimination of the clinical and extended models were similar with a summary AUC of 0.70 (95% CI 0.65–0.74) for the clinical model and a summary AUC of 0.71 (95% CI 0.68–0.75) for the extended model (S1 and S2 Figs). Calibration was satisfactory for both models with a summary calibration slope of 0.83 (95% CI 0.67–0.99) for the clinical model and a summary calibration slope of 0.84 (95% CI 0.71–0.97) for the extended model (S3 and S4 Figs).

**Table 3. Clinical prediction models with clinical characteristics of the index visit to predict the presence of a serious illness at the time of revisit.**

| | | Odds Ratio (95% CI) | Odds Ratio (95% CI) |
|---|---|---|---|
| | | *Clinical model* | *Extended model* |
| *Day and time of arrival* | Weekday evenings | 1.01 (0.85–1.19) | 1.04 (0.88–1.23) |
| | Weekday nights | 1.22 (0.98–1.53) | 1.26 (1.01–1.58) |
| | Weekend days | 1.20 (1.00–1.44) | 1.21 (1.01–1.46) |
| | Weekend evenings | 1.06 (0.84–1.34) | 1.10 (0.86–1.39) |
| | Weekend nights | 1.29 (0.95–1.73) | 1.32 (0.98–1.78) |
| | Weekdays | *reference* | *reference* |
| *Season* | Winter | 1.17 (0.99–1.39) | 1.17 (0.99–1.39) |
| | Spring | 1.15 (0.96–1.37) | 1.14 (0.96–1.37) |
| | Summer | 0.99 (0.82–1.19) | 0.96 (0.80–1.16) |
| | Autumn | *reference* | *reference* |
| *Age* | <1 year | 2.21 (1.74–2.80) | 2.66 (2.09–3.39) |
| | 1 - <2 years | 1.36 (1.05–1.76) | 1.63 (1.26–2.12) |
| | 2 - <5 years | 1.16 (0.91–1.47) | 1.33 (1.05–1.69) |
| | 5 - <12 years | 1.01 (0.80–1.27) | 1.08 (0.86–1.36) |
| | 12–16 years | *reference* | *reference* |
| *Gender* | Female | 0.94 (0.82–1.06) | 0.93 (0.82–1.06) |
| *Presenting problem* | Shortness of breath | 1.89 (1.33–2.67) | 1.91 (1.35–2.70) |
| | ENT problems | 0.83 (0.53–1.30) | 0.85 (0.54–1.34) |
| | Gastro-intestinal problems | 2.61 (1.91–3.57) | 2.31 (1.69–3.17) |
| | Neurological problem | 2.32 (1.57–3.43) | 2.20 (1.49–3.26) |
| | Unwell child | 1.77 (1.29–2.42) | 1.64 (1.19–2.25) |
| | Urological problems | 1.26 (0.74–2.15) | 1.07 (0.63–1.84) |
| | Rash | 0.82 (0.53–1.29) | 0.89 (0.57–1.40) |
| | Abscess and soft tissue infection | 2.21 (1.31–3.71) | 2.22 (1.32–3.73) |
| | Wounds | 0.48 (0.33–0.69) | 0.43 (0.29–0.63) |
| | Trauma | 0.36 (0.18–0.69) | 0.41 (0.21–0.79) |
| | Other | *reference* | *reference* |
| *Triage urgency* | Emergent / very urgent | 1.92 (1.52–2.43) | 1.68 (1.33–2.12) |
| | urgent | 1.83 (1.57–2.13) | 1.67 (1.43–1.94) |
| | Standard / non-urgent | *reference* | *reference* |
| *Tachycardia* | present | 1.31 (1.10–1.55) | 1.29 (1.08–1.53) |
| *Tachypnoea* | present | 1.18 (1.00–1.41) | 1.18 (0.99–1.40) |
| *Temperature* | > = 38.0 degrees Celsius | 0.97 (0.80–1.18) | 0.92 (0.75–1.11) |
| *Oxygen saturations* | Oxygen saturation <94% | 1.50 (0.95–2.37) | 1.43 (0.90–2.25) |
| *Level of consciousness* | Reduced | 1.54 (0.93–2.55) | 1.40 (0.85–2.31) |
| *Laboratory tests* | Any | - | 1.58 (1.35–1.84) |
| *Imaging* | Any | - | 1.61 (1.36–1.92) |
| *IV medication or fluids* | Any | - | 1.89 (1.52–2.35) |

CI confidence interval; ENT: Ear nose and throat; iv: intravenous.

## Derivation of the final clinical and extended prediction models

Next, we combined all cohorts and derived a clinical model predicting the risk of a revisit with serious illness based on clinical characteristics (Table 3; model coefficients are presented in S4 Table). 4,433 children (5%) had a risk of > = 3%, which was useful for ruling in a revisit with serious illness, with positive likelihood ratio 4.41 (95% CI 3.87–5.011) and specificity 0.96

**Table 4. Performance of the prediction models in the cross-validation cohorts.**

| | | Clinical model | | | Extended model | | |
|---|---|---|---|---|---|---|---|
| Hospital | *Cohort name* | Calibration Coefficient (95% CI) | intercept | AUC (95% CI) | Calibration Coefficient (95% CI) | intercept | AUC (95% CI) |
| Erasmus MC, Rotterdam, NL | Erasmus cohort | 0.88 (0.85–0.91) | 0.06 | 0.71 (0.70–0.72) | 0.86 (0.84–0.89) | 0.07 | 0.73 (0.72–0.74) |
| Maasstad Hospital, Rotterdam, NL | Maasstad cohort | 0.86 (0.83–0.98) | 0.10 | 0.75 (0.73–0.77) | 0.86 (0.83–0.89) | 0.11 | 0.76 (0.75–0.78) |
| St. Mary's Hospital, London, UK | London cohort | 0.81 (0.77–0.86) | -0.11 | 0.69 (0.67–0.70) | 0.83 (0.79–0.88) | -0.05 | 0.70 (0.68–0.72) |
| Hospital Fernando da Fonseca, Lisbon, Portugal | Lisbon cohort | 1.06 (1.03–1.09) | -0.02 | 0.72 (0.71–0.73) | 1.03 (1.01–1.06) | -0.01 | 0.73 (0.72–0.74) |
| Medizinische Universitaet Wien, Vienna, Austria | Vienna cohort | 0.55 (0.50–0.60) | -0.91 | 0.61 (0.59–0.63) | 0.61 (0.57–0.66) | -0.61 | 0.63 (0.61–0.65) |

AUC: Area Under the receiver operating Characteristic curve; CI: confidence interval.

(95% CI 0.95–0.96) (Table 5). Increasing this high-risk threshold to 4% or above resulted in improved 'rule in' value (positive likelihood ratio 6.23 (95% CI 5.11–7.61) and specificity 0.99 (95% CI 0.98–0.99)), but at the cost of being applicable to fewer children (n = 1,523, 2%). 37,546 (39%) had a risk <0.5%, which was useful for ruling out a revisit with serious illness (negative likelihood ratio 0.30 (95% CI 0.25–0.35), sensitivity 0.88 (95% CI 0.86–0.90)). An extended model had similar overall diagnostic performance for the low risk threshold of 0.5% and high-risk threshold of 4% (Table 5). The extended model improved risk classification for a modest proportion of children (Fig 2). The clinical model based on all cohorts combined had an optimism corrected apparent AUC of 0.73 (95% CI 0.70–0.76); the extended model also including intravenous medications and/or fluids, imaging, and laboratory investigations had improved optimism corrected AUC of 0.75 (95% CI 0.72–0.78; Likelihood Ratio test for overall model performance improvement, p <0.001). Bias corrected calibration showed goodness of fit of both the clinical and extended models (S5 Fig). Variance inflation factors were between 1 and 6 for all predictor variables indicating limited collinearity between predictor variables (S6 Table).

## Discussion

### Principal findings

Young age under two years, high urgency, and type of presenting problem were highly associated with a return visit to the emergency department with a serious illness. Rates of revisits with serious illness varied between the five hospitals, ranging from 0.7 to 2.2%, and were in line with the lower end of previously reported rates [3, 4, 7, 9, 11]. A clinical prediction model, including the variables season, time and day of presentation, age, gender, presenting problem, triage urgency, and vital signs, was able to select a high number of children with a low risk of return visit with serious illness, as well as identify those with the highest risks. Initial derivation and cross-validation of the model showed the robustness of the model in all settings. We developed a digital calculator to enable clinical implementation. For illustration, in our case example shown in Fig 3, a 3 year boy with shortness of breath had a 3.1% predicted risk of having a return visit with a serious illness, which was between the >70th—≦80th centiles (S5 Table). The predicted probabilities for a revisit with serious illness remained relatively low, even for children with an increased risk, mostly as a result of the overall low prevalence (1.0%). Performing laboratory tests and imaging, as well as administration of intravenous medications

**Table 5. Diagnostic performance of the clinical prediction model for revisits with serious illness.**

| Clinical model | Risk threshold | N (%) | Sensitivity | Specificity | PPV | NPV | LR + | LR - |
|---|---|---|---|---|---|---|---|---|
| | ≥0.25% | 83,748 (86%) | 0.97 (0.96–0.98) | 0.14 (0.14–0.14) | 0.01 (0.01–0.01) | 1.00 (1.00–1.00) | 1.13 (1.11–1.14) | 0.22 (0.16–0.32) |
| | ≥0.5% | 59,723 (61%) | 0.88 (0.86–0.90) | 0.39 (0.39–0.39) | 0.02 (0.01–0.02) | 1.00 (1.00–1.00) | 1.45 (1.42–1.48) | 0.30 (0.25–0.35) |
| | ≥1% | 37,385 (38%) | 0.73 (0.70–0.76) | 0.62 (0.62–0.62) | 0.02 (0.02–0.02) | 1.00 (0.99–1.00) | 1.91 (1.84–1.99) | 0.44 (0.40–0.49) |
| | ≥2% | 13,727 (14%) | 0.38 (0.35–0.42) | 0.86 (0.86–0.86) | 0.03 (0.03–0.03) | 0.99 (0.99–0.99) | 2.78 (2.57–3.01) | 0.71 (0.68–0.75) |
| | ≥3% | 4,433 (5%) | 0.19 (0.17–0.22) | 0.96 (0.95–0.96) | 0.04 (0.04–0.05) | 0.99 (0.99–0.99) | 4.41 (3.87–5.01) | 0.84 (0.82–0.87) |
| | ≥4% | 1,523 (2%) | 0.09 (0.08–0.11) | 0.99 (0.98–0.99) | 0.06 (0.05–0.08) | 0.99 (0.99–0.99) | 6.23 (5.11–7.61) | 0.92 (0.90–0.94) |
| | ≥5% | 428 (0.4%) | 0.03 (0.02–0.04) | 1.00 (1.00–1.00) | 0.06 (0.04–0.09) | 0.99 (0.99–0.99) | 6.38 (4.34–9.37) | 0.98 (0.97–0.99) |
| Extended model with interventions | ≥0.25 | 82,323 (85%) | 0.97 (0.96–0.98) | 0.01 (0.01–0.01) | 0.01 (0.01–0.01) | 1.00 (1.00–1.00) | 1.15 (1.13–1.16) | 0.20 (0.14–0.29) |
| | ≥0.5% | 59,769 (61%) | 0.90 (0.88–0.91) | 0.39 (0.39–0.39) | 0.02 (0.01–0.02) | 1.00 (1.00–1.00) | 1.47 (1.44–1.50) | 0.27 (0.22–0.32) |
| | ≥1% | 35,412 (36%) | 0.73 (0.70–0.76) | 0.64 (0.64–0.64) | 0.02 (0.02–0.02) | 1.00 (1.00–1.00) | 2.03 (1.96–2.11) | 0.42 (0.38–0.46) |
| | ≥2% | 12,802 (13%) | 0.42 (0.39–0.46) | 0.87 (0.87–0.87) | 0.03 (0.03–0.04) | 0.99 (0.99–0.99) | 3.30 (3.07–3.55) | 0.66 (0.63–0.70) |
| | ≥3% | 5,335 (5%) | 0.23 (0.20–0.26) | 0.95 (0.95–0.95) | 0.04 (0.04–0.05) | 0.99 (0.99–0.99) | 4.35 (3.87–4.88) | 0.81 (0.79–0.84) |
| | ≥4% | 2,373 (2%) | 0.12 (0.10–0.14) | 0.99 (0.98–0.99) | 0.05 (0.04–0.06) | 0.99 (0.99–0.99) | 5.09 (4.29–6.05) | 0.90 (0.88–0.92) |
| | ≥5% | 1,133 (1%) | 0.07 (0.06–0.09) | 0.99 (0.99–0.99) | 0.06 (0.05–0.08) | 0.99 (0.99–0.99) | 6.52 (5.19–8.20) | 0.94 (0.92–0.95) |

Diagnostic accuracy of different risk thresholds of the prediction models. The individual risk predictions are based on average risk predictions across the 25 imputed datasets; n = 97,269 children out of n = 98,561 had risk predictions. Overall prevalence of serious illness was 1.0%.

PPV: positive predictive value; NPV negative predictive value; LR likelihood ratio.

were all associated with an increased risk of a revisit with serious illness and improved overall risk prediction, even as the diagnostic performance for low and high-risk threshold were similar for the clinical and extended model.

## Comparison to previous literature

Reasons for children to represent with a serious illness after initial ED visit are diverse. One might be concerned about medical errors in the decision making process during the first visit [27, 28]. However, in one study, return visits resulting in hospital admission were rarely the result of poor quality of ED care at first visit [29]. Another study also suggested that only a minority of return visits with serious illness were the result of a failure to make an appropriate diagnosis, a failure to instigate appropriate treatment, or a failure to investigate fully, and that the large majority of these children with a revisit resembled disease progression [30]. For example, in meningococcal disease is it well known that children might present with non-specific signs and symptoms at first, with the severity of the meningococcal septicaemia only becoming manifest at a later stage of the disease [31]. There might also be parental factors involved, such as a refusal to be admitted in first instance, non-adherence to prescribed

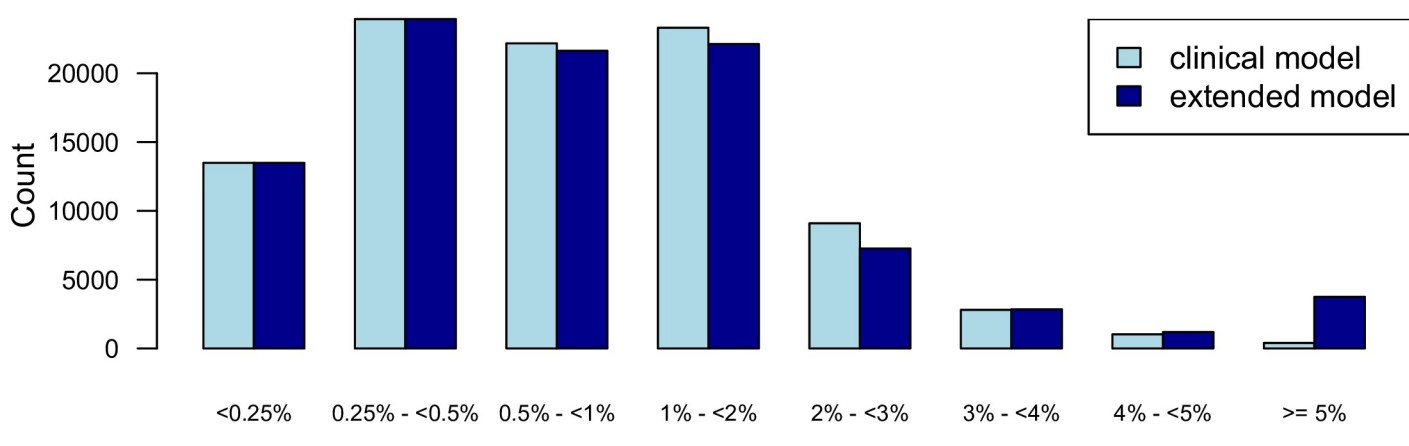

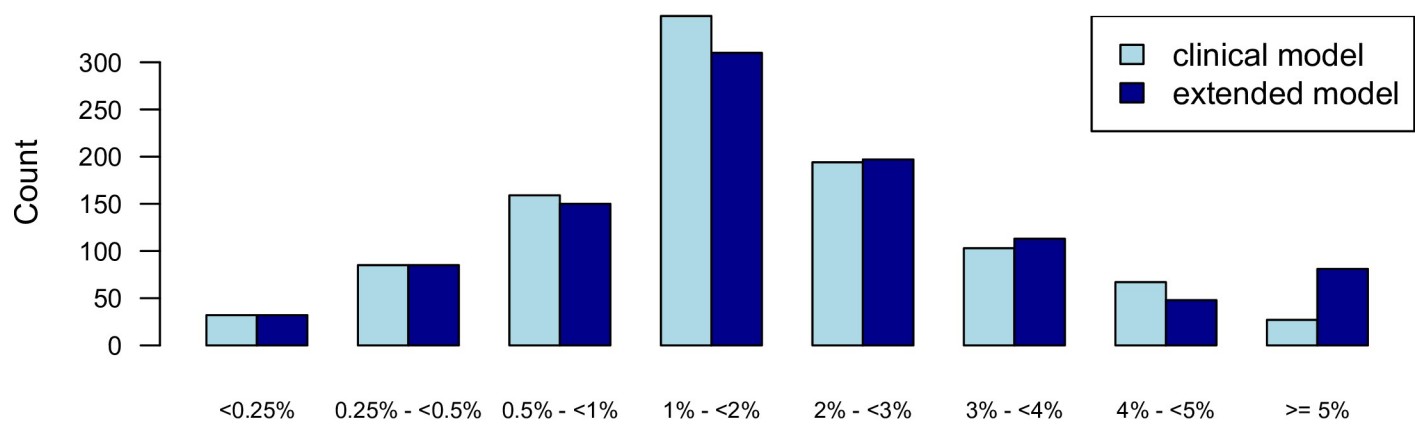

**Fig 2. Barplots of the predicted probabilities of the two prediction models.** Barplots showing the distributions of the predicted risks for the clinical model (top) and extended model (bottom).

therapy at home, or a failure to attend a follow up appointment [30]. This all contributes to the importance of empowering parents to provide informed monitoring at home, and providing them with clear instructions on when to return to the ED [32].

More advanced modelling techniques have been used to provide clinicians with clinical tools to predict return visits to the ED [1, 33, 34]. However, to our knowledge, none focused on the risk of return visits with serious illness, and none presented a tool ready for clinical implementation. From the outset, we felt it was more important to understand the group of children at risk for clinical deterioration, rather than preventing avoidable revisits, even though both have significant implications for appropriate use of scarce resources.

| Clinical scenario: |
|---|
| *You are the attending paediatrician in your local Emergency department of a district general hospital. It is a Wednesday afternoon at 3 pm in springtime. You have just seen a 3 year old boy with shortness of breath. His triage urgency was 'Urgent'. On arrival he had a heart rate of 113 bpm and a respiratory rate of 36 bpm, with mild increased work of breathing and oxygen saturations of 93% in room air. You treated him with inhaled short-acting bronchodilators. After a period of observation he is clinically well and you intend to discharge him with safety netting and home treatment advice.* |
| **Clinical question:** |
| *What is the risk that this child will re-attend the emergency department with a serious illness?* |
| **Use of the digital calculator [http://www.rimon.nl/rvc/]** |
| *The digital calculator gives the following risks:* <br> *[clinical model: 3.1%; extended model 2.2%]* |
| **Advice:** |
| *Using the clinical model, this boy has a **[3.1%]** probability that he will return to the ED with a serious illness. This risk belongs to the **[>70th - ≦80th]** centile of all children seen in your emergency department that are being discharged home.* <br><br> *The average risk for all children that are being discharged home, based on the data used for the development of the model was 1.0%.* |

**Fig 3. Clinical scenario and digital calculator.** A clinical scenario with calculation of the predicted risks for a return visit with serious illness after discharge from the emergency department, using the digital calculator at *http://www.rimon.nl/rvc/*.

One of the difficulties was capturing the vast spectrum of problems clinicians are confronted with in the ED in a single tool [16]. For certain conditions, such as children at risk of dehydration or children at risk for severe infection, specific clinical features were identified predictive of clinical deterioration [35, 36]. Although those predictors might be important, they only apply to a select proportion of the children visiting the ED, and this will need to be positioned against the convenience of a 'one-fits-all' tool. Our model assigned higher risks of serious illness at revisit to children presenting with specific problems, such as shortness of breath, gastro-intestinal problems, and neurological problems.

Tachycardia has been shown to be associated with an increased risk of a revisit, as well as with revisits with serious illness, yet we did not find evidence of this [11, 37]. We did not include heart rate at the time of discharge; however, Daymont et al. looked at this and only found limited additional value for different ways of including heart rate as a predictor of return visits with serious illness [37]. The role of other vital signs as predictors of serious illness needs further exploring.

Some variables, such as social economic status, ethnicity, language fluency, and health literacy were previously shown to be associated with revisits as well as revisits with serious illness, yet were not included in the model [2–4, 7, 10]. Studies had inconsistent and sometimes contradicting risk profiles for these variables [36]. Crowding of EDs has previously been associated with both an increased as well as a reduced risk [2–4]. Moreover, Akenroye et al. found that ED crowding was an increased risk for revisits, but not for a revisit with admission to hospital

[3], highlighting that risk factors for return visits differ from risk factors for revisits with serious illness. Notably, ethnicity and variables derived from the use of postal codes, such as deprivation scores, cannot routinely be used in most European countries without explicit informed consent and hence were not included in our models.

## Implications for clinical practice and future research

The limited variability in calibration slopes testifies that this model can be used in a diverse range of clinical settings [25]. The differing underlying incidence of serious revisits, as also reflected in the intercept values of the cross-validation studies ('calibration-in-the-large', Table 4), indicate that the local incidence should be taken into account when interpreting the risk predictions. If the local incidence is known, one could adjust our model for local use by adjusting the intercept of the model. Knowing which children are at increased risk for clinical deterioration following discharge from ED can contribute to more tailored, patient-specific follow-up [15]. How this follow-up should look will be highly setting dependent, with some institutions having the capacity to organise planned face to face review appointments in ED or by an ambulatory care team, and in other settings this might be done via telephonic consultation or with the involvement of primary care. Many factors contribute to the development of serious illness, including natural disease progression and treatment compliance. By implementing tailored safety netting and follow-up, this will help early detection of children with progressing disease, preventing delayed presentations after an initial health care visit, as well as ensuring treatments are continued at home, and are given in a timely and appropriate manner. Examples of the latter could include antibiotic tolerance in young children, appropriate inhaler technique of bronchodilators, and reinforcement of hydration strategies. More senior clinicians will likely have an experienced feel for those patients at highest risk for deterioration. Our models enable this for more junior physicians, who see the large majority of unwell children in acute care facilities, and for settings with no dedicated paediatric emergency care team. Also, our prediction model can complement decision making in acute paediatric conditions where a watchful waiting approach is justified and a return visit can be interpreted as a marker of good clinical care, such as in children with non-specific abdominal pain deemed to be at low risk for appendicitis at first assessment. Importantly, any intervention with individualised follow-up or targeted safety netting will need to be studied carefully, in combination with cost-effectiveness analyses, as it might inadvertently have unwanted effects. One study showed an increase in the rate of return visits following the introduction of telephone calls after ED discharge [38]. In future, studies should also look into the reasons for revisiting the ED, such as parental factors, potential of medical errors, and natural disease course, as they will be useful in fine-tuning the specific interventions for safety netting and follow-up. In addition, the potential for machine learning techniques to improve our prediction model should be explored.

## Strengths

This is the first study including a large cohort of children using a prospective multicentre design with data from five different European EDs. The design allowed for standardised, near complete data on predictors known to be related with urgency systems and severity scores. We opted for including clinical variables with minimal interuser variability that were all readily available in routine clinical practice. For example, the reliability and validity of the MTS triage urgency classification was shown previously [39]. We also converted our understanding of determinants of revisits with serious illness into a clinical tool that can be used in clinical practice. We included the clinical variables independent of their statistical contribution to the

model to ensure validity of the model and enhance clinical uptake. Although one might argue that the discriminative ability of the model is only moderate, the tool is able to identify a large number of children who can be discharged safely, and also identifies a select and well-defined group of patients that can be targeted for more tailored safety netting advice and follow-up strategies.

## Limitations

One of the limitations was the over-representation of patients from the larger of the five hospitals, and we adjusted for this by including the hospitals as a categorical variable whilst developing and validating the prediction models. Generally, findings across the sites were similar (data not presented), as was also confirmed by the cross-validation analyses. Furthermore, one can argue the accuracy of the outcome definition for serious illness; however, defining serious illness by the need for hospitalisation at revisit was least subjective to a doctor's impression, and with limited variability between hospitals [40]. We performed a secondary analysis with a broader outcome definition of serious illness, including those who underwent lifesaving interventions or had triage urgency 'emergent' at revisit. This yielded only 16 additional cases, which did not affect our overall findings, supporting the validity of our outcome definition. Despite our large sample size, it did not allow for a separate model for children admitted to PICU at revisit (S2 Table). In addition, we lack data to show the impact of revisits to other EDs than the one originally visited, and future studies will need to provide detailed insight into where children with a serious illness re-present during an episode of acute illness. As our study hospitals have an important regional referral function with limited admission capacity, hospital admission was warranted only to provide therapeutic of supportive treatment that would not have been able in the community or by other health care providers in the large majority of cases. The role of other types of admission, such as for social reasons, will need to be studied in future. Also, only a relatively small number of variables were included resulting in a low explained variance of our models, meaning future studies should explore the incremental predictive value of additional variables, such as social economic status and language. We did not consider language fluency, as reliably determining parental or carer's language proficiency was not feasible. Another important candidate predictor will be co-morbidity, which was an important predictor of return visits with serious illness in three out of five cohorts with data on co-morbidity available. Furthermore, the validity and applicability of our prediction models still needs to be proven outside Europe and in different health systems., specifically in those without primary care or universal health care coverage. Lastly, the data used for this study are from the period prior to the COVID-19 pandemic. The COVID-19 pandemic has impacted local healthcare delivery systems globally, and future validation studies will need to study long term impact of the current COVID-19 pandemic.

## Conclusion

We developed a prediction model and a digital clinical calculator that can aid physicians identifying those children at highest and lowest risks for developing a serious illness after initial discharge from the ED, allowing for more targeted safety netting advice and follow-up.

## Supporting information

**S1 Appendix. Sample size estimation.**
(PDF)

**S1 Table. Definitions of presenting symptoms and lifesaving interventions.**
(PDF)

**S2 Table. Characteristics of the index visit of children admitted to PICU at the time of revisit.**
(PDF)

**S3 Table. Clinical prediction models for the three hospitals with data on co-morbidity available.**
(PDF)

**S4 Table. Regression coefficients of the final clinical prediction models.**
(PDF)

**S5 Table. Percentiles of risk.**
(PDF)

**S6 Table. Variance inflation factors.**
(PDF)

**S1 Fig. Area under the receiver operating curve (AUC) for validation cohorts.**
(PDF)

**S2 Fig. Forest plots for cross validation area under the receiver operating curves (AUC).**
(PDF)

**S3 Fig. Forest plots for cross validation calibration slopes.**
(PDF)

**S4 Fig. Calibration plots for cross-validation cohorts.**
(PDF)

**S5 Fig. Bias corrected calibration plots.**
(PDF)

## Acknowledgments

We would like to thank Rikke Jorgensen for her dedication as a research nurse to the study at the St Mary's hospital–Imperial College NHS Healthcare Trust. Similarly, we would like to acknowledge Pinky Rose Espina, paediatric resident at Medizinische Universitaet Wien, Vienna, for her contributions.

## Author Contributions

**Conceptualization:** Ruud G. Nijman, Dorine H. Borensztajn, Joany M. Zachariasse, Paulo Freitas, Susanne Greber-Platzer, Frank J. Smit, Claudio F. Alves, Johan van der Lei, Ewout W. Steyerberg, Ian K. Maconochie, Henriette A. Moll.

**Data curation:** Ruud G. Nijman, Dorine H. Borensztajn, Joany M. Zachariasse, Paulo Freitas, Susanne Greber-Platzer, Claudio F. Alves, Johan van der Lei, Ewout W. Steyerberg, Ian K. Maconochie, Henriette A. Moll.

**Formal analysis:** Ruud G. Nijman, Dorine H. Borensztajn, Joany M. Zachariasse, Carine Hajema, Johan van der Lei, Ewout W. Steyerberg, Ian K. Maconochie, Henriette A. Moll.

**Funding acquisition:** Ruud G. Nijman, Paulo Freitas, Susanne Greber-Platzer, Frank J. Smit, Claudio F. Alves, Johan van der Lei, Ewout W. Steyerberg, Ian K. Maconochie, Henriette A. Moll.

**Investigation:** Ruud G. Nijman, Joany M. Zachariasse, Carine Hajema, Frank J. Smit, Johan van der Lei, Ewout W. Steyerberg, Henriette A. Moll.

**Methodology:** Ruud G. Nijman, Dorine H. Borensztajn, Joany M. Zachariasse, Carine Hajema, Paulo Freitas, Susanne Greber-Platzer, Claudio F. Alves, Johan van der Lei, Ewout W. Steyerberg, Ian K. Maconochie, Henriette A. Moll.

**Project administration:** Carine Hajema, Paulo Freitas, Susanne Greber-Platzer, Frank J. Smit, Ewout W. Steyerberg, Ian K. Maconochie, Henriette A. Moll.

**Resources:** Joany M. Zachariasse, Paulo Freitas, Susanne Greber-Platzer, Frank J. Smit, Claudio F. Alves, Johan van der Lei, Ewout W. Steyerberg, Ian K. Maconochie, Henriette A. Moll.

**Software:** Ruud G. Nijman, Dorine H. Borensztajn, Johan van der Lei, Henriette A. Moll.

**Supervision:** Ruud G. Nijman, Dorine H. Borensztajn, Paulo Freitas, Susanne Greber-Platzer, Frank J. Smit, Claudio F. Alves, Johan van der Lei, Ewout W. Steyerberg, Ian K. Maconochie.

**Validation:** Ruud G. Nijman, Ewout W. Steyerberg, Henriette A. Moll.

**Writing – original draft:** Ruud G. Nijman, Dorine H. Borensztajn, Joany M. Zachariasse, Carine Hajema, Ian K. Maconochie, Henriette A. Moll.

**Writing – review & editing:** Ruud G. Nijman, Dorine H. Borensztajn, Joany M. Zachariasse, Carine Hajema, Paulo Freitas, Susanne Greber-Platzer, Frank J. Smit, Claudio F. Alves, Johan van der Lei, Ewout W. Steyerberg, Ian K. Maconochie, Henriette A. Moll.

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
