## [Decision Letter · Decision Letter 0]

7 Apr 2021

PONE-D-20-36121

A clinical prediction model to identify children at risk for revisits with serious illness to the emergency department: a prospective multicentre observational study

PLOS ONE

Dear Dr. Nijman,

Your manuscript "*A clinical prediction model to identify children at risk for revisits with serious illness to the emergency department: a prospective multicentre observational study*" has been assessed by two reviewers. Based on these reports and my assessment as Editor, I am pleased to inform you that it is potentially acceptable for publication in PLoS One Journal once you have carried out some essential revisions suggested by the reviewers. Their reports, together with many other comments, are below.

Thank you for providing the Tripod checklist. When completing the list, please use section and paragraph numbers rather than page numbers.

Did your study have a prospective protocol or analysis plan? Please state this (either way) early in the Methods section. If a prospective analysis plan was used in designing the study, please include the relevant prospectively written document with your revised manuscript as a Supporting information file to be published alongside your study, and cite it in the Methods Section. If no such document exists, please ensure that the Methods section transparently describes when analyses were planned and when/why any data-driven changes to analyses took place.

Best wishes,

Jose Moreira

We look forward to receiving your revised manuscript.

Kind regards,

José Moreira, MD, MSc

Academic Editor

PLOS ONE

Journal Requirements:

4. Please include a caption for all figure.

Reviewers' comments:

Reviewer's Responses to Questions

**Comments to the Author**

1. Is the manuscript technically sound, and do the data support the conclusions?

Reviewer #1: Yes

Reviewer #2: Yes

2. Has the statistical analysis been performed appropriately and rigorously? 

Reviewer #1: Yes

Reviewer #2: Yes

3. Have the authors made all data underlying the findings in their manuscript fully available?

Reviewer #1: Yes

Reviewer #2: Yes

4. Is the manuscript presented in an intelligible fashion and written in standard English?

Reviewer #1: Yes

Reviewer #2: Yes

5. Review Comments to the Author

Reviewer #1: The article is interesting, original and relevant when proposing a prediction model that helps in the pediatric emergency care. It aims to identify children who are more likely probability to revisit ED with serious illness. The article is well written, and the analysis was well conducted. Several points were already answered previously by authors for the other reviewers. However, some explanations about the statistical models are not completely clear yet.

Major comments

1. I do not understand the sentence “Using the 10 cases : 1 predictor rule of thumb, our cohort provided enough power to evaluate approximately 100 predictor variables….(21)” (page 8). I understand that the authors use the “rule” of 10 observations (cases) per predictor variable, but it is controversial, and it is not sufficient to justify a study power. The ideal would be calculate the sample size or power, but I understand the obstacles to calculate it without parameters. I suggest that authors exclude this sentence.

2. The additional analysis with comorbidities in three hospitals seems misplaced or it is not linked to objective and the other analysis. I suggest the authors that consider the need of this inclusion. It is not clear why only the effect of comorbidities is shown in the table S3, while several effects are shown in the table 2. Moreover, the sentence in the results section “Comorbidity”….(S3 appendix 3, page 10) does not highlighted the differences between the values.

3. The term “stepwise approach” (page 8) can cause misunderstandings. It was clarified by the authors in the answers for other reviewer, but this sentence in the text remains unclear. I only suggest that authors excluded the name of the approach (e.g. Secondly, we developed two clinical prediction models using logistic regression: clinical model and extended model. The first model…..).

4. The authors show the explicative (Table 2) and predictive (other tables) models. The role of the explicative model (Table 2) is not clear in the text, I think that it was used to select the factors to the predictive model.

5. It is not clear the reason to not include the goodness-of-fit measures of the logistic models (such as, Hosmer and Lemeshow test and/or residual analysis).

6. Some prediction variables can be correlated (multicollinearity) and the use of VIF measure could be important.

7. It is suggested that the authors include the interpretation of AUC in the methods section.

8. The crude OR would be useful to compare with adjusted OR and to detect modification effects.

9. Why are not the medication effect explained in the paragraph “Characteristics…serious illness” (page 10)?

10. It is important highlighted that the multicovariate models are joint adjusted by all variables, so the interpretation of effects should be clear about this adjustment. For example, the paragraph “Characteristics of the index…serious illness” (page 10) only mention about the significant effects.

11. The guidelines often ask to avoid repeating the values from tables and graphs (ORs, AUCs). It is suggested that the authors try to reduce the use of the values in the text.

12. Although the explanation about discrimination and calibration in the paragraph “Overall, discrimination (summary AUC)…(table3)” (page 11) can be understood, the citations of tables and figures are grouped all together and it is hard to check them. One suggestion is split these citations along the sentences and maybe compare the clinical and extended models in distinct sentences, one about the discrimination and another about the calibration.

13. Some confidence intervals have the same values for the lower and upper limits (e.g. 0.97-0.97, page 11 and in some tables).

14. There is an increase critical to use of p-values in big data, hence some p-values and effects (ORs) can show significance. It is suggested to provide this limitation in the discussion section.

15. The likelihood ratio test was used to interpret p-values of all models, but it is not clear in the methods section.

16. It was not finding the same probabilities (3.1% and 2.2%) with the parameters provided in the example (figure 3). The digital calculator could have some explanations about the variable names and the clinical and extended definitions.

17. It does not seem adequate to interpret the effects and outcome as “risk”, so the authors used logistic models and ORs.

18. A short explanation about the different health systems across sites could be useful in the methods section and/or in the limitations, stressing the type of care in the sites (e.g. universal).

19. Two points could be mentioned in the limitations, the external validity to other countries (eg. not-developed countries, outside Europe and different health systems) and the statistical limitations (use of pvalue to build the regression model).

Minor comments

1. The name of design study (cohort study?), as well the hospitals, would be useful in the “Design, participants and setting” subsection. The abbreviations about hospital are in the page 7, but they are not defined before.

2. Number and name of hospitals could be defined in the “Design, participants and setting” (page 6).

3. Are there some references for the sentence “Data were extracted from the hospital’s electronic systems and checked for completeness, validity and outliers” (page 6)?

4. Some abbreviations are not defined in the first citation in the text (e.g. PICU-Abstract section, AVPU-methods section, EMC, SMH and MUW- page 7).

5. What are the variables specified in the sentence “Only variables with data available in all settings were used in the multivariable regression analysis” (page 6)? The tables do only seem show variables which were included in the modelling. I think this sentence is not necessary. The same happens with sentence “with high number of missings were not considered…” (page 8).

6. It was not cited the use of odds ratio and confidence intervals in the methods section.

7. The term “cross validation” could be used in the methods section to clarify the explanation about “leave-one-out“ approach.

8. Some ORa values are incorrect if we compare with table 2 (page 10).

9. The sentence “First, we developed…cohort once.” (page 11) is more appropriate in the methods section.

10. The sentence “Intercept coefficients ranged…calibration-in-the large” (page 11) seems misplaced. I suggest that it is moved for the beginning or the end of the paragraph.

11. It is suggested to check the guidelines for formatting tables (e.g. lines, definitions of abreviations, etc). For example, it was not defined the clinical and extended model in the table 3.

12. The use of term “multivariable” can bring some misunderstands. Although many publications in epidemiology use this term to refer the adjustment for several covariables, in the statistical this term refers to a multivariate outcome. I recommend change this term, such as multi-covariate logistic model.

13. The term MICE refers to the package number, but is also refers to name of the statistical method, so I recommend that the complete name is included (eg. Imputation was performed using the Multivariate imputation by chained equations in the MICE package….).

14. It is suggested to include the version of R software, not only, the version of the mice package.

15. The figure 2 seems not necessary since the authors already include the information in the table and the text.

16. It is suggested to include labels (eg. a,b,c) for subfigures, which can facilitate the citation in the text.

Reviewer #2: Thank you for the opportunity to review the manuscript. The authors have developed a clinical prediction model to identify children at risk for revisits with serious illness to the emergency department.

Although I am not medically qualified to judge the clinical aspects of the modelling, my overall impression is that the authors should be congratulated for undertaking this important research that will likely have a clinical implication.

However, I have few methodological questions/suggestions to the authors, clarification of which might further help the readers of the manuscript.

1. Adequacy of sample size for model development

Page 8, lines 13-14: Using the 10 cases: 1 predictor rule of thumb, our cohort provided enough power to evaluate approximately 100 predictor variables.

The “rule of thumb” of 10 events for predictor as used by the authors is subject to criticism in many methodological literature. For example, please see the following article: https://pubmed.ncbi.nlm.nih.gov/29966490/

I would also like to direct the authors to the recent methods proposed by Riley et al. that provides a much more robust framework to assess the adequacy of the sample size the authors have used and will allow them to assess the minimum number of “parameters” (not predictors”) can the authors consider for development of a new prognostic model

https://pubmed.ncbi.nlm.nih.gov/30357870/

https://www.bmj.com/content/368/bmj.m441

2. Adequacy of sample size for validation

Page 8, lines 13-14: “Similarly, each cohort had >100 cases to allow for sufficient power for the validation studies.”

Again, If possible, please consider assessing if the validation cohort still meets the new criterion presented in a recent methodological publication:

https://onlinelibrary.wiley.com/doi/full/10.1002/sim.8766

I understand that it might be an extreme request at this stage of revision, as this newly suggested rule didn’t exist at the time the authors began their model development. However, it may be worthwhile to flag this up in the discussion.

 

3. Imputation model

Page 8-9: “Missing data for vital signs (table 2) were imputed using a multiple imputation model including a hospital variable, age, available vital signs, triage urgency, presenting problem, and discharge destination”

From their statement, it appears that the “outcome” (i.e. serious illness defined as hospital admission or PICU admission or death ED after an unplanned revisit within 7 days of the index visit) from the analysis model is not included in the imputation model. It is known that an imputation model that doesn’t include the outcome will lead to biased estimates.

https://pubmed.ncbi.nlm.nih.gov/16980150/

https://pubmed.ncbi.nlm.nih.gov/21225900/

Please clarify if that was the case or not (“a hospital variable” was included and it is not clear if that was the outcome of the target analysis). If outcome was not included, then please consider justifying the reasons for such non-exclusion.

4. Table 3: Performance measures

Are these performance measures optimism-corrected? If it was please clearly indicated or perhaps consider referring these measures as Optimism corrected performance measures in the table of the title.

5. Table 4 and S7 Appendix Table

Are these coefficients adjusted for optimism/shrinkage?

Apologies for asking these if these were corrected for optimism as it was not evident that these methods had been applied from the methods and the results section.

6. PLOS authors have the option to publish the peer review history of their article (what does this mean?). If published, this will include your full peer review and any attached files.

Reviewer #1: No

Reviewer #2: No

---

## [Author Response · Author response to Decision Letter 0]

26 May 2021

PONE-D-20-36121

A clinical prediction model to identify children at risk for revisits with serious illness to the emergency department: a prospective multicentre observational study

PLOS ONE

Dear Dr. Nijman,

- Your manuscript "A clinical prediction model to identify children at risk for revisits with serious illness to the emergency department: a prospective multicentre observational study" has been assessed by two reviewers. Based on these reports and my assessment as Editor, I am pleased to inform you that it is potentially acceptable for publication in PLoS One Journal once you have carried out some essential revisions suggested by the reviewers. Their reports, together with many other comments, are below.

We thank the editor for giving us the opportunity to address some of the remaining issues related to our manuscript. We greatly appreciate the supportive comments to our manuscript, which we believe describes important and novel research. In this revised version of the manuscript, we have attempted to incorporate the reviewers’ and editorial feedback, and we feel this has significantly improved the manuscript. Hopefully we addressed all comments sufficiently, and we are looking forward to discussing any remaining questions. 

- Thank you for providing the Tripod checklist. When completing the list, please use section and paragraph numbers rather than page numbers.

We have added section and paragraphs to the Tripod checklist, using the lay out of the revised manuscript.

- Did your study have a prospective protocol or analysis plan? Please state this (either way) early in the Methods section. If a prospective analysis plan was used in designing the study, please include the relevant prospectively written document with your revised manuscript as a Supporting information file to be published alongside your study, and cite it in the Methods Section. If no such document exists, please ensure that the Methods section transparently describes when analyses were planned and when/why any data-driven changes to analyses took place.

The analyses presented in this manuscript were embedded in the TRiAGE study. We referenced another recent study detailing the original study objectives and analyses plans in our methods section (Zachariasse et al. Lancet Child Adolesc Health, 2020). 

The full reference for Zachariasse et al: 

Zachariasse JM, Nieboer D, Maconochie IK, et al. Development and validation of a Paediatric Early Warning Score for use in the emergency department: a multicentre study. Lancet Child Adolesc Heal [Internet]. 2020 Aug 1;4(8):583–91. 

We have stated that this study is a secondary analysis of existing data in both the abstract and the Methods, p.6 line 9. The data analyses were planned and executed after completion of the original data collection, and we have added a sentence to the methods section to clarify this: 

Methods, section: design, participants and setting, p.6 line 14:

Data analyses were planned and executed following completion of the original data collection.

Importantly, all variables needed for the objectives and analyses for the current study were part of the original data collection, with no additional data collection needed. Moreover, the objectives of this study are closely related to the aims of the TRiAGE study, and the study group agreed that the current study proposal would fit well within the wider remit of the study. We have previously included the protocol of the TRiAGE study on original submission to PLOS Medicine; however, we feel that referencing to the existing publications would be more useful than including the protocol as supplementary file as the protocol doesn’t explicitly specify the research question or outcome measure proposed in this study. 

Another recent study by Zachariasse was published in PLOS One using the same study protocol and TRIAGE study dataset: 

Zachariasse JM, Maconochie IK, Nijman RG, et al. Improving the prioritization of children at the emergency department: Updating the Manchester Triage System using vital signs. PLoS One [Internet]. 2021 Feb 9;16:e0246324. 

We added this reference to the manuscript. 

Journal Requirements:

We have attempted to comply with all the journal’s style requirements. Please let us know if there are any style issues that need addressing. 

We have updated our reference list to include a reference supporting our updated estimation of the required sample size, as suggested by both reviewers. 

We have included captions for the supporting information files at the end of the revised manuscript.

4. Please include a caption for all figure.

We have also added these to manuscript. 

We will make all data necessary to replicate this study’s findings publicly available without restriction at the time of publication. A DOI with access to the full dataset used for this study will be provided upon acceptance. 

Reviewer #1: The article is interesting, original and relevant when proposing a prediction model that helps in the pediatric emergency care. It aims to identify children who are more likely probability to revisit ED with serious illness. The article is well written, and the analysis was well conducted. Several points were already answered previously by authors for the other reviewers. However, some explanations about the statistical models are not completely clear yet.

We thank the reviewer for these supportive comments, and we have hopefully addressed any remaining issues in this revised version of the manuscript. 

Major comments

1. I do not understand the sentence “Using the 10 cases : 1 predictor rule of thumb, our cohort provided enough power to evaluate approximately 100 predictor variables….(21)” (page 8). I understand that the authors use the “rule” of 10 observations (cases) per predictor variable, but it is controversial, and it is not sufficient to justify a study power. The ideal would be calculate the sample size or power, but I understand the obstacles to calculate it without parameters. I suggest that authors exclude this sentence.

We agree with the reviewer that the method used for estimating power can be considered controversial and now outdated. In this revised manuscript we applied the method for sample size estimation as suggested by Riley et al., which we believe would be a more appropriate manner for estimating the required sample size. 

We added to Methods, section statistical analysis, p.8 line 23:

Our cohort provided sufficient power for both the derivation and cross-validation studies, estimating the required sample using the recommended approach by Riley et al. (S3 Appendix)

We added appropriate references in this section of the text. 

We added a separate appendix (S3 Appendix) to support our sample size calculations applying the method suggested by Riley et al. We used the existing prediction model of Hu et al. , albeit using a different outcome (ie: any return visits), for the expected Cox-Schnell R2 model performance. As we included pre-selected variables only, we used the included variables (and variable categories) as the number of parameters used for power estimation. 

Importantly, we believe that we have ample power to perform the derivation and cross validation of the prediction models. 

2. The additional analysis with comorbidities in three hospitals seems misplaced or it is not linked to objective and the other analysis. I suggest the authors that consider the need of this inclusion. It is not clear why only the effect of comorbidities is shown in the table S3, while several effects are shown in the table 2. Moreover, the sentence in the results section “Comorbidity”….(S3 appendix 3, page 10) does not highlighted the differences between the values.

We thank the reviewer for this valuable comment. 

We have discussed the need for describing the association of co-morbidity with return visits and serious illness with the study group extensively. What we tried to show in this manuscript, and by including data in the manuscript and appendix, that co-morbidity is associated with return visits with serious illness. Unfortunately, we did not have these data available for two of our cohorts. To avoid being criticised for not exploring this variable, we included the analyses for the three cohorts for which we had these data available. 

Altogether, we have reported on the association between comorbidity and return visits with serious illness, but we lacked the data to include in the overall prediction models. We hope to have positioned these data to support future research updating the prediction models with this variable, as argued in Discussion, section on Limitation, p. 18, line 12-13. 

We still feel that, even though not included in the prediction models, it will be worth reporting on our findings of the association between comorbidity and return visits with serious illness. We initially opted to present the adjusted Odds Ratios, without presenting the aORs for the other model variables. However, as suggested by the reviewer, we have now included the effects of the other variables in the model as well (S4 Table). 

3. The term “stepwise approach” (page 8) can cause misunderstandings. It was clarified by the authors in the answers for other reviewer, but this sentence in the text remains unclear. I only suggest that authors excluded the name of the approach (e.g. Secondly, we developed two clinical prediction models using logistic regression: clinical model and extended model. The first model…..).

We thank the reviewer for this suggestion. We have now excluded the sentence about the stepwise approach from the methods altogether, as we agree with the reviewer that this terminology was confusing. 

4. The authors show the explicative (Table 2) and predictive (other tables) models. The role of the explicative model (Table 2) is not clear in the text, I think that it was used to select the factors to the predictive model.

To clarify, we did not perform any variable selection for the variables included in the prediction models (Table 3), based on p-values (or similar) in the original Table 2. All variables for the final prediction models were pre-selected, based on both existing literature and/or availability of data (i.e.: some variables had high numbers of missings, such as blood pressure, and others were not available, such as socio-economic status, as explained in the discussion). 

Based on additional arguments made by the reviewers, we have now changed Table 2. 1) We excluded any p-values; and 2) we changed the adjusted ORs to crude unadjusted ORs. The adjusted ORs can be found in Table 3 as part of the presentation of the clinical prediction models. Hopefully, we have now removed any uncertainty about explicative vs predictive models. 

We feel that the current Table 2 helps showing completeness and distribution of the data, or in some instances the high level of missings such as for the BP variable. Table 2 also provides the numbers per hospital, not included in tables elsewhere.

 5. It is not clear the reason to not include the goodness-of-fit measures of the logistic models (such as, Hosmer and Lemeshow test and/or residual analysis).

We initially did not include goodness-of-fit measures, such as the Hosmer Lemeshow test, and their usefulness is often debated for clinical prediction models. Our preferred method to show ‘goodness of fit’ is a bias corrected calibration plot, based on 200 bootstrap iterations, which we have included in S11 Fig. 

We added to Methods, section. Statistical analyses, p.10 line 8: 

We performed bias corrected calibration with 200 bootstrapping iterations to demonstrate goodness of fit for the final overall prediction models

We added to Results, section clinical prediction model, p.13, line 1:

Bias corrected calibration showed goodness of fit of both the clinical and extended models (S11 Fig).

For completeness, we also performed HL goodness-of-fit-tests for both the clinical model (p=0.47) and the extended models (p=0.83); we have not included this in the main text of the manuscript. 

6. Some prediction variables can be correlated (multicollinearity) and the use of VIF measure could be important.

We agree with the reviewer that it is important to consider collinearity between predictor variables in models exploring associations between predictors and outcomes; when dealing with prediction models this fortunately doesn’t seem to be as much of a concern. We added the variance inflation factors as a supplementary table, with all VIF reassuringly between 1 and <6. 

We added to Methods, section. Statistical analyses, p.10 line 10: 

Variance inflation factors were calculated to assess collinearity between predictor variables.

We added to Results, section clinical prediction model, p.13, line 2-3:

Variance inflation factors were between 1 and 6 for all predictor variables indicating limited collinearity between predictor variables (S12 table).

7. It is suggested that the authors include the interpretation of AUC in the methods section.

We have included this in the Methods, section statistical analysis, p.9 line 23-25:

The AUC is used to estimate the ability to discriminate between patients with and without a return visit with serious illness, with values ranging between 0.5 (indiscriminate test) and 1 (optimal test). 

8. The crude OR would be useful to compare with adjusted OR and to detect modification effects.

At this suggestion of the reviewer, we have now changed the odds ratios in Table 2 to crude unadjusted odds ratios. Although one can sometimes argue about the added value of presenting crude unadjusted Odds Ratios, it is interesting to see how much the associations change after adjusting for confounding variables in our study. 

9. Why are not the medication effect explained in the paragraph “Characteristics…serious illness” (page 10)?

We have changed this section of the results. In the revised version of the manuscript, we have 1) mentioned imaging, laboratory tests and intravenous fluids/medications and their association with the risk of a return visit with serious illness; 2) removed the actual aORs as to avoid duplication from tables, in response to another issue raised by the reviewers. 

10. It is important highlighted that the multicovariate models are joint adjusted by all variables, so the interpretation of effects should be clear about this adjustment. For example, the paragraph “Characteristics of the index…serious illness” (page 10) only mention about the significant effects.

We agree with the importance of this comment, and that the associations are fully dependent on the variables adjusted for. This is clearly illustrated by the differences in the unadjusted (Table 2) and adjusted Odds Ratios (Table 3). We hope that the current revised version of the manuscript with changes to the wording of the Results section, Table 2, and Table 3 has made this clearer. 

11. The guidelines often ask to avoid repeating the values from tables and graphs (ORs, AUCs). It is suggested that the authors try to reduce the use of the values in the text.

We value this issue raised by the reviewer. We reviewed this carefully and made some changes, particularly in the results section. For example, we removed the aORs in the section on the Characteristics of return visits with serious illness, as to avoid duplication from tables. 

12. Although the explanation about discrimination and calibration in the paragraph “Overall, discrimination (summary AUC)…(table3)” (page 11) can be understood, the citations of tables and figures are grouped all together and it is hard to check them. One suggestion is split these citations along the sentences and maybe compare the clinical and extended models in distinct sentences, one about the discrimination and another about the calibration.

We thank the reviewer for this suggestion. We have rewritten this paragraph (Results, section on Cross-validation of the clinical and extended prediction models) to first compare the AUCs, and then the calibration for both the models, with clearer reference to the relevant supplemental files. 

13. Some confidence intervals have the same values for the lower and upper limits (e.g. 0.97-0.97, page 11 and in some tables).

We have checked all these confidence intervals, and can confirm that these are accurate; the narrow intervals are a result of the considerable cohort sizes. 

 14. There is an increase critical to use of p-values in big data, hence some p-values and effects (ORs) can show significance. It is suggested to provide this limitation in the discussion section.

We fully agree with the reviewer on this point. In response to this comment, we have now removed the p-values in Table 2 and (current) Table 3, with only the crude unadjusted ORs (Table 2) and adjusted Odds Ratios (Table 3) remaining. 

15. The likelihood ratio test was used to interpret p-values of all models, but it is not clear in the methods section.

We have changed the wording in the methods section to include this: 

We have included this in the Methods, section Statistical analysis, p.9 line 8:

The overall performance of the extended model was compared with the clinical model using the likelihood ratio test.

16. It was not finding the same probabilities (3.1% and 2.2%) with the parameters provided in the example (figure 3). The digital calculator could have some explanations about the variable names and the clinical and extended definitions.

We appreciate that the reviewer noticed this difference. The digital calculator was still using different model coefficients from an earlier test model. We will update the coefficients, and finalise the wording used, upon acceptance of the manuscript. 

17. It does not seem adequate to interpret the effects and outcome as “risk”, so the authors used logistic models and ORs.

We apologise for not fully understanding this point. We use a mixture of methods to express risks and diagnostic performance of the models, such as odds ratios, AUC, sensitivity, specificity and likelihood ratios, to show 1) performance and robustness of the prediction models, 2) role of the separate predictor variables associated with the outcome, and 3) help physicians with their clinical decision making. Hopefully this is now sufficiently clear in this revised manuscript. 

18. A short explanation about the different health systems across sites could be useful in the methods section and/or in the limitations, stressing the type of care in the sites (e.g. universal).

Across the sites included in our study insurance coverage, access to urgent and emergency care, and availability of primary care were similar. All countries in our study have primary care with general practitioners as well as primary care paediatricians (Austria, Portugal) available, free of charge. To summarise the health systems: 

a. UK: a publicly funded National Health Service with open access, free of charge, to primary care, and urgent and emergency care. 

b. The Netherlands has universal health care, with all adults are required to have a basic health insurance; children are insured via their parents. Primary care is freely accessible; there is an own risk for most other healthcare (385 euros per year per adult in 2020).

c. Portugal has universal health care coverage via the publicly financed national health service. 

d. Austria has a two-tier health care system in which all individuals receive publicly funded care and additional private care is available. 

We added to the Methods, section on design, setting, and participants, p.6, line 16-19:

Universal health care coverage was available across the participating sites, with similar and free of charge access to urgent and emergency care, and primary care. Primary care with either general practitioners and/or primary care paediatricians was available in all the countries in our study

We added to Discussion, section on Limitation, p.18, line 14-16

Furthermore, the validity and applicability of our prediction models still needs to be proven outside Europe and in different health systems., specifically in those without primary care or universal health care coverage.

19. Two points could be mentioned in the limitations, the external validity to other countries (eg. not-developed countries, outside Europe and different health systems) and the statistical limitations (use of pvalue to build the regression model).

We have included a comment on the generalizability of the model outside West European settings. To clarify, we did not select variables using p values, as also addressed earlier, and hence have not included this as a limitation. 

We added to Discussion, section on Limitation, p.18, line 14-16

Furthermore, the validity and applicability of our prediction models still needs to be proven outside Europe and in different health systems., specifically in those without primary care or universal health care coverage.

Minor comments

1. The name of design study (cohort study?), as well the hospitals, would be useful in the “Design, participants and setting” subsection. The abbreviations about hospital are in the page 7, but they are not defined before.

We added the design (ie: prospective, observational, multicentre, cohort study) to the Section on Setting, design, and participants.

2. Number and name of hospitals could be defined in the “Design, participants and setting” (page 6).

We added the names of the five participating hospitals to the text of the Section on Setting, design, and participants, p.6.

3. Are there some references for the sentence “Data were extracted from the hospital’s electronic systems and checked for completeness, validity and outliers” (page 6)?

We have not included any specific references for this data quality monitoring and checking process. If the reviewer has any specific concerns, or needs specific parts of this process to be addressed into more details, we would be happy to do so. 

4. Some abbreviations are not defined in the first citation in the text (e.g. PICU-Abstract section, AVPU-methods section, EMC, SMH and MUW- page 7).

Thank you for noticing this: we have now adjusted this in the revised manuscript. 

 5. What are the variables specified in the sentence “Only variables with data available in all settings were used in the multivariable regression analysis” (page 6)? The tables do only seem show variables which were included in the modelling. I think this sentence is not necessary. The same happens with sentence “with high number of missings were not considered…” (page 8).

We thank the reviewer for this suggestion. We have removed these two sentences from the Methods. To make clear why certain variables were not used for the prediction models, we added to the Methods, section on predictor variables, p.8 line 4-7:

Blood pressure, capillary refill and lifesaving interventions were not used for prediction modelling owing to low number of values measured in children across all institutions (blood pressure available for 8%; capillary refill for 47%); and/or <10 cases with abnormal values on initial ED attendance or return visit (Table 2).

We separately explained that co-morbidity was not available in all cohorts. 

6. It was not cited the use of odds ratio and confidence intervals in the methods section.

We have included a statement on the use of odds ratios and 95% confidence intervals in the methods section. 

7. The term “cross validation” could be used in the methods section to clarify the explanation about “leave-one-out“ approach.

We added this to the Methods, section on statistical analyses, p. 9, line 15-18:

We initially derived the prediction model in four of the five cohorts, and then cross-validated the model in the fifth cohort that was left out. We repeated this “leave-one-out“ approach as a method for cross-validating the prediction model four times with each of the five cohorts serving as an independent validation cohort once. 

8. Some ORa values are incorrect if we compare with table 2 (page 10).

We thank the reviewer for pointing these inconsistencies out. We have carefully reviewed these and adjusted as appropriate in the revised manuscript; however, in response to a previous comment we decided to leave the aORs out of this section of the Results, referring only to Table 3. 

9. The sentence “First, we developed…cohort once.” (page 11) is more appropriate in the methods section.

We thank the reviewer for this suggestion, and we have removed this section from the Results. We believe the explanation in the Methods should now be sufficiently clear in describing the methods applied for our cross-validation studies. 

10. The sentence “Intercept coefficients ranged…calibration-in-the large” (page 11) seems misplaced. I suggest that it is moved for the beginning or the end of the paragraph.

In response to an earlier comment, we have rewritten this paragraph, with the modifications making this section clearer and hopefully addressing this comment in full. 

11. It is suggested to check the guidelines for formatting tables (e.g. lines, definitions of abreviations, etc). For example, it was not defined the clinical and extended model in the table 3.

We have updated our revised manuscript to adhere to PLOS style regulations. 

12. The use of term “multivariable” can bring some misunderstands. Although many publications in epidemiology use this term to refer the adjustment for several covariables, in the statistical this term refers to a multivariate outcome. I recommend change this term, such as multi-covariate logistic model.

We thank the reviewer for this suggestion, and we have changed this accordingly. We assume this terminology is now also in line with PLOS journal terminology recommendations. 

13. The term MICE refers to the package number, but is also refers to name of the statistical method, so I recommend that the complete name is included (eg. Imputation was performed using the Multivariate imputation by chained equations in the MICE package….).

We thank the reviewer for this suggestion and we have changed this accordingly. 

14. It is suggested to include the version of R software, not only, the version of the mice package.

As we now used additional packages in R, we have slightly rewritten the methods on the software used; it now reads: Methods, Section on statistical analyses, p. 10, line 12-14.

We used R statistical software version 4.0.0 for all our analyses, including use of the rms, metaphor, MICE, pmsampsize, pROC and epiR packages.

15. The figure 2 seems not necessary since the authors already include the information in the table and the text.

We agree that most of the information contained in Figure 2 is also available in the text and Table 5, and hence we understand that the reviewer brings up this issue. However, we felt that including this plot would help the reader with visualising the distributions of the predicted risks. Some of the numbers presented in tables and text might not intuitively give the reader a full impression of the spectra of predicted risks, especially as for example the calibration plots are in the supplemental files rather than the main manuscripts, and we felt this graph helps to convey that important information. If the editor feels that the figure indeed duplicates information, we are willing to move this graph to the supplemental files. 

16. It is suggested to include labels (eg. a,b,c) for subfigures, which can facilitate the citation in the text.

We will gladly be guided by any editorial suggestions on doing this for any figures in the main manuscript or in the supplemental files. 

Reviewer #2: Thank you for the opportunity to review the manuscript. The authors have developed a clinical prediction model to identify children at risk for revisits with serious illness to the emergency department. Although I am not medically qualified to judge the clinical aspects of the modelling, my overall impression is that the authors should be congratulated for undertaking this important research that will likely have a clinical implication. However, I have few methodological questions/suggestions to the authors, clarification of which might further help the readers of the manuscript.

We thank the reviewer for these supportive comments. We have now hopefully addressed all remaining issues below. 

1. Adequacy of sample size for model development

Page 8, lines 13-14: Using the 10 cases: 1 predictor rule of thumb, our cohort provided enough power to evaluate approximately 100 predictor variables. The “rule of thumb” of 10 events for predictor as used by the authors is subject to criticism in many methodological literature. For example, please see the following article: https://pubmed.ncbi.nlm.nih.gov/29966490/

I would also like to direct the authors to the recent methods proposed by Riley et al. that provides a much more robust framework to assess the adequacy of the sample size the authors have used and will allow them to assess the minimum number of “parameters” (not predictors”) can the authors consider for development of a new prognostic model

https://pubmed.ncbi.nlm.nih.gov/30357870/

https://www.bmj.com/content/368/bmj.m441

We agree with the reviewer that the method suggested by Riley et al. has become the preferred method used for estimating power for the derivation and validation of prediction models. We thank the reviewer for pointing us in the direction of some of the more recent publications. In this revised manuscript we have now applied the method for sample size estimation as suggested by Riley et al. 

We added to Methods, section statistical analysis, p.8 line 23-25:

Our cohort provided sufficient power for both the derivation and cross-validation studies, estimating the required sample using the recommended approach by Riley et al. (S3 Appendix)

We added appropriate references in this section of the text. 

We added a separate appendix (S3 Appendix) to support our sample size calculations applying the method suggested by Riley et al. We used the existing prediction model of Hu et al. , albeit using a different outcome (ie: any return visits), for the expected Cox-Schnell R2 model performance. As we included pre-selected variables only, we used the included variables (and variable categories) as the number of parameters used for power estimation. 

Importantly, we believe that we have ample power to perform the derivation and cross validation of the prediction models. 

2. Adequacy of sample size for validation

Page 8, lines 13-14: “Similarly, each cohort had >100 cases to allow for sufficient power for the validation studies.

Again, If possible, please consider assessing if the validation cohort still meets the new criterion presented in a recent methodological publication: https://onlinelibrary.wiley.com/doi/full/10.1002/sim.8766

I understand that it might be an extreme request at this stage of revision, as this newly suggested rule didn’t exist at the time the authors began their model development. However, it may be worthwhile to flag this up in the discussion.

Hopefully we answered this issue above, and we have made changes to the revised manuscript. Similar principles apply for the estimation of sample size requirements for the derivation as well as the validation of the models. The model performance of our models, as measured with Cox Schnell R2, was slightly lower than that of Hu et al. (as derived from AUC), which was used to guide the initial power estimation. Still, we feel we have enough power for the cross-validation studies, even taking into consideration that the cohort sizes of our 5 cohorts vary somewhat, with a range of numbers of cases and non-cases (as seen in Table 2). 

3. Imputation model

Page 8-9: “Missing data for vital signs (table 2) were imputed using a multiple imputation model including a hospital variable, age, available vital signs, triage urgency, presenting problem, and discharge destination” From their statement, it appears that the “outcome” (i.e. serious illness defined as hospital admission or PICU admission or death ED after an unplanned revisit within 7 days of the index visit) from the analysis model is not included in the imputation model. It is known that an imputation model that doesn’t include the outcome will lead to biased estimates.

https://pubmed.ncbi.nlm.nih.gov/16980150/

https://pubmed.ncbi.nlm.nih.gov/21225900/

Please clarify if that was the case or not (“a hospital variable” was included and it is not clear if that was the outcome of the target analysis). If outcome was not included, then please consider justifying the reasons for such non-exclusion.

We thank the reviewer for discussing this issue. We used a broad description of the imputation process in the manuscript, rather than a precise formula. Reassuringly, we did include the outcome for each of the visits under the term ‘discharge destination’, and hence included the outcome in our imputation process. Importantly, we opted to use the same imputed datasets for all data analyses associated with the TRiAGE study, and we did not impute the data for this study separately. For example, this means that the whole TRiAGE study population was used for imputation including those admitted after their first presentation, who are excluded from further analyses in this study (as per flowchart in Figure 1), adding to the precision of the estimates.

4. Table 3: Performance measures

Are these performance measures optimism-corrected? If it was please clearly indicated or perhaps consider referring these measures as Optimism corrected performance measures in the table of the title. 

We decided to use uncorrected estimates for the cross-validation studies, as shown in Table 4 (previous Table 3); moreover, as the random effects modelling already provides conservative estimates of the summary AUC, calibration-in-the-large and calibration slope. We strived to present these summary statistics as the model performance, for example by mentioning these in the abstract, rather than the apparent AUC. 

We did use the optimism corrected AUC for the apparent AUC of the clinical and extended models. We now made this clearer by adding this to Methods and Results. 

Methods, section on Statistical Analyses, p.10 line 7-8:

An apparent optimism corrected AUC was calculated by means of bootstrapping (500 iterations).

Result, section on Derivation of the final clinical and extended models , p.12 line 22-24:

The clinical model based on all cohorts combined had an optimism corrected apparent AUC of 0.73 (95% CI 0.70–0.76); the extended model also including intravenous medications and/or fluids, imaging, and laboratory investigations had improved optimism corrected AUC of 0.75 (95% CI 0.72–0.78)

5. Table 4 and S7 Appendix Table

Are these coefficients adjusted for optimism/shrinkage?

Apologies for asking these if these were corrected for optimism as it was not evident that these methods had been applied from the methods and the results section.

We thank the reviewer for asking about these details of our modelling approach. 

We did not apply shrinkage during the cross-validation studies. We explored the need for shrinkage of the coefficients of the final model, but we found a shrinkage factor of only 0.03. As this shrinkage factor, based on 500 bootstrap iterations of the final clinical and extended models, was so minor, we opted to use the unadjusted coefficients. We hope the reviewer agrees with us that this was an acceptable decision.

---

## [Editor Report · Decision Letter 1]

28 Jun 2021

A clinical prediction model to identify children at risk for revisits with serious illness to the emergency department: a prospective multicentre observational study

PONE-D-20-36121R1

Dear Dr. Nijman,

Your manuscript has now been formally accepted for publication in *PLoS One*. Please see important details concerning the publication process below.

Your efforts during the process of revision are acknowledged and I hope you also are pleased with the final result.

We appreciate being able to publish your work and look forward to seeing your paper online as soon as possible.

Kind regards,

José Moreira

Academic Editor

PLOS ONE
---

## [Editor Report · Acceptance letter]

8 Jul 2021

PONE-D-20-36121R1 

A clinical prediction model to identify children at risk for revisits with serious illness to the emergency department: a prospective multicentre observational study 

Dear Dr. Nijman:

I'm pleased to inform you that your manuscript has been deemed suitable for publication in PLOS ONE. Congratulations! Your manuscript is now with our production department. 

Kind regards, 

on behalf of

Dr. José Moreira 

Academic Editor

PLOS ONE